# Chronic Experimental Model of TNBS-Induced Colitis to Study Inflammatory Bowel Disease

**DOI:** 10.3390/ijms23094739

**Published:** 2022-04-25

**Authors:** Inês Silva, João Solas, Rui Pinto, Vanessa Mateus

**Affiliations:** 1H&TRC–Health and Technology Research Center, ESTeSL–Escola Superior de Tecnologia da Saúde de Lisboa, Instituto Politécnico de Lisboa, 1990-096 Lisbon, Portugal; ines.silva@estesl.ipl.pt (I.S.); joaoslas96@gmail.com (J.S.); 2iMed.ULisboa, Faculdade de Farmácia, Universidade de Lisboa, 1649-003 Lisbon, Portugal; 3Joaquim Chaves Saúde, Laboratório de Análises Clínicas, 1495-069 Algés, Portugal; rapinto@ff.ulisboa.pt

**Keywords:** inflammatory bowel disease, TNBS-induced colitis, chronic animal model

## Abstract

Background: Inflammatory bowel disease (IBD) is a world healthcare problem. In order to evaluate the effect of new pharmacological approaches for IBD, we aim to develop and validate chronic trinitrobenzene sulfonic acid (TNBS)-induced colitis in mice. Methods: Experimental colitis was induced by the rectal administration of multiple doses of TNBS in female CD-1 mice. The protocol was performed with six experimental groups, depending on the TNBS administration frequency, and two control groups (sham and ethanol groups). Results: The survival rate was 73.3% in the first three weeks and, from week 4 until the end of the experimental protocol, the mice’s survival remained unaltered at 70.9%. Fecal hemoglobin presented a progressive increase until week 4 (5.8 ± 0.3 µmol Hg/g feces, *p* < 0.0001) compared with the ethanol group, with no statistical differences to week 6. The highest level of tumor necrosis factor-α was observed on week 3; however, after week 4, a slight decrease in tumor necrosis factor-α concentration was verified, and the level was maintained until week 6 (71.3 ± 3.3 pg/mL and 72.7 ± 3.6 pg/mL, respectively). Conclusions: These findings allowed the verification of a stable pattern of clinical and inflammation signs after week 4, suggesting that the chronic model of TNBS-induced colitis develops in 4 weeks.

## 1. Introduction

Inflammatory bowel disease (IBD) comprises Crohn’s disease (CD) and ulcerative colitis (UC) [1]. CD and UC are chronic and relapsing inflammatory conditions of the gastrointestinal tract that have distinct pathological and clinical characteristics. The evolution of IBD follows the advancement of society [2,3]. It is almost a global disease, affecting all ages, including the pediatric population [4]. Several treatments are currently available for the treatment of IBD, though none of them reverses the underlying pathogenic mechanism of this disease [5,6].

Animal models of IBD remain essential for a proper understanding of histopathological change in the gastrointestinal tract and play a key role in the development of new pharmacological approaches [7]. The trinitrobenzene sulfonic acid (TNBS)-induced model, in particular, is a commonly used model of IBD since it is capable of reproducing CD in humans [8,9,10]. Additionally, the TNBS-induced model can mimic the acute and chronic stages of IBD [11,12]. TNBS is a haptenizing agent that stimulates a delayed-type hypersensitivity immune response, driving colitis in susceptible mouse strains [13,14,15,16,17]. The chemical is dissolved in ethanol, enabling the interaction of TNBS with colon tissue proteins. The ethanol breaks the mucosal barrier, allowing TNBS to penetrate into the bowel wall [16]. TNBS administration induces transmural colitis that is driven by a Th1-mediated immune response [13,14,17] that is characterized by the infiltration of CD4 cells, neutrophils, and macrophages into the lamina propria and the secretion of cytokines [14] The most common cytokines are tumor necrosis factor (TNF)-α and interleukin (IL)-12. Since the protocols using the TNBS-induced colitis model are not standardized, a systematic review [18] developed by our research group concludes that the chronic TNBS-induced colitis model can be obtained with multiple TNBS administrations. 

In the literature, there is no consensus about the induction method, and several original articles have been published with different ways to induce a chronic model of TNBS-induced colitis using different doses, numbers of TNBS administrations, strains, genders, and ages of mice. By this point of view, the main objective of this study is to develop and validate a chronic TNBS-induced colitis in mice in order to evaluate the effect of new pharmacological approaches for IBD. The advantage of this chronic model compared to acute models is that the latter may provide only limited information about the pathogenesis of human IBDs, as the chemical injury to the epithelial barrier leads to self-limiting inflammation rather than chronic disease [19]. Our research group has previously developed preclinical studies in an acute TNBS-induced colitis model [20,21,22]. However, as cited by Bilsborough and colleagues (2021), since IBD is a chronic disease, the development of a standardized and validated induction method for chronic colitis is useful for studying new pharmacological approaches [23]. Our findings allow us to conclude that TNBS-induced chronic colitis should be developed in 4 weeks, providing a chronic intestinal inflammation model.

## 2. Results

### 2.1. Clinical Signs

For six weeks, the mice were observed daily for body weight, stool consistency, and morbidity. Between weeks 2 and 4, the mice presented an alteration of intestinal motility that was characterized by soft stools and moderate morbidity. On the other hand, after week 4, mice presented an apparent recovery. No alterations were identified in the control groups.

Concerning body weight, all groups demonstrated a very similar curve in the register of body weight during the experiment (Figure 1). Until week 4, the majority of the TNBS groups showed a progressive increase in body weight. After six administrations, the T6 group animals increased 14.2 ± 2.7% from their initial weight. At the end of the experimental period, the ethanol group gained 14.7 ± 2.3% of its initial weight. No significant differences were observed between the groups.

In this experimental procedure, the mortality rate is important to regulate the TNBS dose since the TNBS should induce the disease without inducing a high mortality rate. In this way, the survival curve was recorded, and it demonstrates a decrease in the survival rate in the first three weeks to approximately 73.3% (Figure 2). After week 4, the mice’s survival maintained the same value until the end of the experimental protocol, approximately 70.9%.

### 2.2. Macroscopic Assessment of Colitis

Macroscopically, the colons were observed and scored for gross morphological damage, according to Morris et al., 1989 [24]. The maximal damage in the colon was observed with two administrations of TNBS (T2 group), with a mean score of 2.7 ± 0.7 (*p* < 0.0001 compared with the ethanol group), corresponding to a linear ulcer with inflammation at one site. After week 2, the gross morphology damage decreased and stabilized from week 4. In the T4 group, the attributed score was 0.9 ± 0.1 (*p* < 0.0001 compared with the T2 group), corresponding only to localized hyperemia without ulcers. The ethanol group presented a score of 0, with no damage registered (Figure 3).

### 2.3. Biochemical Markers

Fecal hemoglobin allows the evaluation of the intensity of the hemorrhagic focus (Figure 4). After week 2, the fecal hemoglobin progressively increased until week 6. A significant increase (*p* < 0.01) was observed when comparing the fecal hemoglobin of the T3 group (4.0 ± 0.2 µmol Hg/g feces) with the T4 group (5.8 ± 0.3 µmol Hg/g feces). However, when comparing the fecal hemoglobin of the T4 group (5.8 ± 0.3 µmol Hg/g feces) with the T6 group (7.5 ± 0.5 µmol Hg/g feces), no statistical differences were observed.

Due to its essential role in intestinal homeostasis, The ALP concentration in the blood was evaluated (Figure 5). In general, the ALP levels observed in the TNBS groups were higher than those observed in the control groups, with a maximum ALP level in the T5 group of 58.5 ± 2.2 U/L (*p* < 0.0001 compared with the ethanol group). After week 3, the maintenance of ALP values can be observed, with 39.7 ± 2.4 U/L in the T4 group and 44.4 ± 2.3 U/L in the T6 group and no statistical significance. 

### 2.4. Pro and Anti-Inflammatory Cytokine Levels

This animal model showed significant production of TNF-α, a pro-inflammatory cytokine, after TNBS administration (Figure 6). All TNBS groups presented higher levels of TNF-α than the control groups, except on week 2, where only a slight increase was observed. The T2 group presented 51.4 ± 2.9 pg/mL of TNF-α, while the ethanol group presented 39.8 ± 1.5 pg/mL, without statistical significance. The highest level of TNF-α was observed on week 3 (87.3 ± 13.1 pg/mL). However, on week 4, the T4 group verified a slight decrease to 71.3 ± 3.3 pg/mL, which indicates maintenance of the values over time when compared with the T6 group (72.7 ± 3.6 pg/mL), without statistical significance being observed. Compared with the ethanol group, the T4 group presented statistically significant differences (*p* < 0.0001).

IL-10 plays a central role in the mucosal immune system, inhibiting pro-inflammatory cytokine synthesis. IL-10 concentrations decreased in week 2. However, serum levels progressively increased until week 6 (Figure 7). Comparing the T3 group with the T6 group, the IL-10 concentration progressed from 44.6 ± 3.7 pg/mL to 68.2 ± 1.8 pg/mL, respectively (*p* < 0.0001). However, the T4 group (57.3 ± 3.8 pg/mL) had no significant differences from the T6 group (68.2 ± 1.8 pg/mL).

### 2.5. Histopathological Features

The histopathological analysis allowed the evaluation of colonic injury based on inflammatory cell infiltration and tissue damage.

The colons presented a severe infiltration of inflammatory cells, foci of ulceration with necrosis, and tissue disruption in the T1, T2, and T3 groups (Figure 8). The inflammatory infiltrates were mostly present in the mucosa and submucosa, but some animals presented transmural infiltrates with extension to the mesentery. The T2 group showed the most severe phenotype. In the T4, T5, and T6 groups, the level of inflammatory cell infiltration and areas of epithelial ulceration and tissue disruption decreased.

The colitis severity of each group was calculated by summing individual lesions. The T1 and T4 groups presented the lowest histopathological score, showing inflammation without fibrosis or tissue loss. The T2 group exhibited the most severe score, with tissue loss, epithelial lesion, and inflammation (Figure 9).

## 3. Discussion

The TNBS groups presented several clinical manifestations, including alterations in intestinal motility, characterized by soft stools and diarrhea, and moderate morbidity, which is consistent with the literature [25,26]. The ethanol and sham groups remained free of alterations. The peak of the clinical signs occurred two weeks after induction and was followed by a partial recovery [24,27,28]. The fact that the peak was observed in week 2 is compatible with an acute model. Moreover, in the acute model developed by our research group, it was found that the acute lesions occurred 4 to 5 days after induction. After this period, the model becomes chronic. This was also confirmed with this experimental protocol. The model only worsened in the second week because the mice were subjected to a second administration of TNBS and, once again, very aggressive values occurred 4 to 5 days after this induction. However, from the third week, the mice appeared to show resistance and began to recover on some parameters. From week 4 to week 6, the model remained stable, presenting chronic inflammation.

Concerning body weight monitoring, colitic mice presented a progressive increase in body weight throughout the experimental procedure, with a 10% increase in body weight since the induction day. Mice of the T6 group had a weight gain, after six administrations, consistent with previously reported results [26,29,30,31,32]. However, three weeks after the beginning of the experiment, the TNBS groups showed a decrease in body weight. This decrease may be related to the fragility of the mice following three TNBS administrations. However, a recovery in body weight occurred over the following weeks. This may be consistent with the fact that after the fourth administration mice gain resistance to TNBS.

In addition, around 73% of the mice survived in the first three weeks of the protocol, followed by a stabilization of the survival rate from week four until the end of the study. These results are coincident with the peak of morbidity observed in this study and with the peak of the disease, as described by some authors [24,28,29]. On the other hand, the survival rate indicates that the dose and frequency of TNBS administrations are optimal to induce the disease while minimizing the mortality rate. A TNBS dose reduction could be considered to reduce mortality; however, the disease induction would be compromised.

The peak of colon damage was observed in week 2, corresponding to the linear ulcer with inflammation at one site. This score passed to 1 with “localized hyperemia, but no ulcers” in weeks 4, 5, and 6. On the other hand, the ethanol and sham groups present a score of 0, with no colonic damage. Once again, we can observe a stabilization of the model from week 4. Consistent with the literature, mice from the TNBS groups should present the colon with marked hyperemia and ulcers [33,34,35].

After week 2, the fecal hemoglobin progressively increased until week 6, with a significant increase between the third and the fourth week (*p* < 0.01). As well as the parameters analyzed above, fecal hemoglobin exhibits a stable pattern of TNBS-induced colitis from the fourth administration. These results seem to show the presence of hemorrhagic ulcers and are in accordance with the results obtained by our research group in the acute colitis model [20,21,22].

The TNBS groups presented the highest values of total serum ALP concentration compared to the control groups, suggesting that the increase in ALP levels observed in the TNBS groups was caused by the induction of intestinal injury with TNBS. These results are consistent with other studies [20,21,22,36]. Increased ALP values were maintained from the fourth week, demonstrating again a stabilization of the model from week 4. Comparing the results of the blood concentration of ALP from the ethanol and the sham groups, a slight increase in ALP levels in the ethanol group was observed. The use of ethanol as a TNBS vehicle aims to induce changes in intestinal permeability, enabling the translocation of TNBS into the submucosal layer, causing colon damage [37,38]. The increased values of ALP in the ethanol group confirm this damage.

TNF-α is a pro-inflammatory cytokine that is produced during the innate immune response of IBD [37]. The literature indicates that increased values of this pro-inflammatory cytokine are related to the pathogenesis of IBD [38,39,40]. Our results confirm this tendency. A significant increase in the levels of TNF-α was observed in the colon. Specifically, this cytokine recorded a peak in the third week. However, on week 4 there was a slight decrease and maintenance until week 6, indicating the possible onset of the chronic phase of this inflammatory disease. In acute inflammation, the values of pro-inflammatory cytokines are very high [20,21,22]. Conversely, in chronic inflammation, the values are only slightly increased. In contrast, anti-inflammatory cytokines act as a brake on this process, preventing an exacerbated response and possibly producing undesirable effects on the inflammation itself and the healing process [41]. The presence of IL-10 was measured in parallel to pro-inflammatory cytokines since it plays a central role in the mucosal immune system [42]. A decrease in IL-10 was observed in week 2. However, after this week, the production of IL-10 progressively increased until the sixth week. Normally, when the values of TNF-α increased, IL-10 decreased. However, from the second week, the values of IL-10 increased. This is probably an intrinsic compensatory mechanism that is trying to solve the inflammation by increasing this anti-inflammatory cytokine. The literature confirms this tendency of the immune system to restore homeostasis, balancing the values of biochemical markers of inflammation [43,44].

Moreover, the histological examination of the colon showed a severe infiltration of inflammatory cells in the mucosa that was associated with ulceration and tissue disruption. Our results are consistent with the literature, which also reports damage of the mucosal architecture simultaneously with a thickening of the colon wall, ulcers, and extensive inflammatory cell infiltration in the colonic mucosa of colitic mice [27,45,46].

## 4. Materials and Methods

### 4.1. Material

2,4,6-Trinitrobenzene sulfonic acid (TNBS 5%) was purchased from Sigma-Aldrich Chemical. Ketamine (Ketamidor^®^ 100 mg/mL, Lisbon, Portugal) was purchase from Richter Pharma. Xylazine (Sedoxylan^®^ 20 mg/mL, Lisbon, Portugal) was purchased from Dechra. An ADVIA^®^ kit was purchased from Siemens Healthcare Diagnostics (Madrid, Spain). An ELISA assay kit for TNF-α measurement was obtained from Hycult Biotechnology.

### 4.2. Animals 

Female CD-1 mice, weighing 20–30 g and aged 6–10 weeks, were obtained from the Institute of Hygiene and Tropical Medicine. The animals were housed in standard polypropylene cages with ad libitum access to food and water in the bioterium of the Faculty of Pharmacy of the University of Lisbon. The mice were kept at 18–23 °C and 40–60% humidity in a controlled 12 h light/dark cycle. All animal care and experimental procedures were performed in accordance with the internationally accepted principles for laboratory animal use and care, Directive 2010/63/EU, transposed to the Portuguese legislation by Directive Law 113/2013. The experiment was approved by the institutional animal ethics committee (ORBEA) of the Faculty of Pharmacy of the University of Lisbon, 3/2020.

### 4.3. Trinitrobenzene Sulfonic Acid-Induced Colitis

The mice were left unfed for 24 h before the induction day. On the induction day (day 0), the mice were anesthetized with an intraperitoneal (IP) injection of ketamine 100 mg/Kg + xylazine 10 mg/Kg (40 µL/mice), and a catheter was carefully inserted into the colon until the tip was 4 cm proximal to the anus. Then, 100 µL of 1% TNBS in 50% ethanol was administered, and the mice were kept in a Trendelenburg position for 1 min. This procedure was repeated weekly, for a period of 5 weeks. On weeks 1, 2, 3, 4, 5, and 6, depending on the experimental TNBS group, the mice were anesthetized, and blood samples were collected by cardiac puncture. The mice were sacrificed by cervical dislocation. The necropsy was initiated with a midline incision into the abdomen. The colon was separated from the surrounding tissues and removed.

### 4.4. Experimental Groups

The mice were categorized into eight groups: six TNBS groups, depending on the number of TNBS administrations, and two control groups. The TNBS groups were: the T1 group (*n* = 15) that received one TNBS administration at week 0; the T2 group (*n* = 15) that received two TNBS administrations at weeks 0 and 1; the T3 group (*n* = 15) that received three TNBS administrations at weeks 0, 1, and 2; the T4 group (*n* = 15) that received four TNBS administrations at weeks 0, 1, 2, and 3; the T5 group (*n* = 15) that received five TNBS administrations at weeks 0, 1, 2, 3, and 4; and the T6 group (*n* = 15) that received six TNBS administrations at weeks 0, 1, 2, 3, 4, and 5. The control groups were the ethanol (E) and sham (S) groups; the E group (*n* = 15) received 100 µL of 50% ethanol (TNBS vehicle) and the S group (*n* = 15) received 100 µL of saline solution. 

### 4.5. Monitoring of Clinical Signs

After induction, the animals were observed daily for the monitoring of body weight, stool consistency, morbidity, and mortality.

### 4.6. Macroscopic Assessment of Colitis

A macroscopic assessment of TNBS-induced colitis was performed by using the criteria for the scoring of gross morphologic damage, as previously described by Morris et al., 1989 [24]. The score was attributed based on the presence of hyperemia, ulcers, and inflammation and the number of sites with ulceration and/or inflammation and its extension.

### 4.7. Biochemical Markers

Serum from the collected blood samples was separated by centrifugation at 3600 rpm for 15 min. A serum analysis was conducted in order to evaluate alkaline phosphatase (ALP) using an automated clinical chemistry analyzer (ADVIA^®^ 1200, Madrid, Spain). Fecal hemoglobin was evaluated using a quantitative method by immunoturbidimetry (Krom Systems).

### 4.8. Measurement of Cytokines

The pro-inflammatory cytokine TNF-α and the anti-inflammatory cytokine IL-10 were measured and expressed in pg/mL. Colonic tissue samples from each animal were weighed and homogenized in phosphate buffer (Ultra-turrax T25, 13,500 rev/min, twice for 30 s). Afterward, samples were centrifuged (15,000 rpm for 15 min at 4 °C). The aliquots of the supernatant were conserved at −20 °C until use. A spectrophotometric measurement of the cytokine level was performed at 450 nm (ELISA kit Quantikine, Hycult Biotechnology, Abingdon, UK).

### 4.9. Histopathological Analysis

The histopathology was carried out by an independent histopathologist from the Gulbenkian Institute of Science who was blinded to the groups. Colon samples were fixed in 10% phosphate-buffered formalin, processed routinely for paraffin embedding, sectioned at 5 µm, and stained with hematoxylin and eosin. To increase the possibility of detecting fibrosis, Masson’s trichrome staining was used. Sections of the distal colon were evaluated based on adapted criteria of Seamons and colleagues (2013) [47]. The histopathological score of lesions was partially scored (0–4 increasing in severity) with some parameters, namely: (1) the presence of tissue loss/necrosis; (2) the severity of the mucosal epithelial lesion; (3) inflammation; (4) extent 1—the percentage of the intestine affected in any manner; and (5) extent 2—the percentage of intestine affected by the most severe lesion. The colitis severity was calculated by summing the individual lesions and the extent scores, promoting a final colitis score (max score = 20).

### 4.10. Statistical Analysis 

The results were expressed as the mean ± SEM of N observations, where N represents the number of animals analyzed. Data analysis was performed using SPSS software (version 26.0). The results were analyzed by a one-way ANOVA to determine the statistical significance between the TNBS and control groups. For multiple comparisons, Tukey’s post hoc test was used. A *p*-value of less than 0.05 was considered significant.

## 5. Conclusions

A TNBS-induced colitis model was monitored for 6 weeks. A scheme of multiple TNBS administrations was performed since the aim was to achieve a chronic pattern of induced colitis and to identify the week in which the damage becomes chronic.

Clinical manifestations of chronic colitis usually peak within 2 weeks and may be followed by partial recovery or death. These results were expected and are compatible with the correct induction of colitis [26]. Our findings allow us to conclude that TNBS-induced chronic colitis should be developed in 4 weeks, providing a chronic intestinal inflammation model. Indeed, the parameters under evaluation, such as clinical manifestations; biochemical markers, including fecal hemoglobin and pro- and anti-inflammatory cytokine levels; and macroscopic evaluation, corroborate that the chronic illness pattern is observed from week 4 after the induction. Some mice died during the early days of the study, possibly because they did not resist the aggravation of the disease in its acute phase. On the other hand, the remaining mice resisted and progressed to the chronic phase of the disease, showing the same symptoms but more lightly.

## Figures and Tables

**Figure 1 ijms-23-04739-f001:**
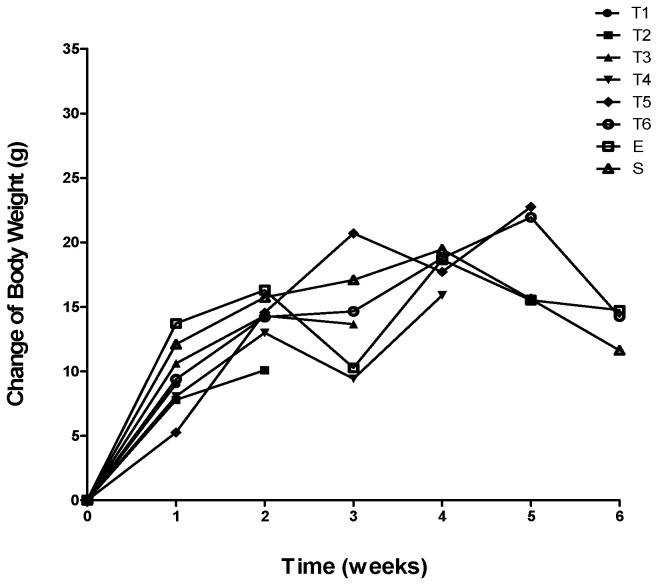
Change in body weight during the development of TNBS-induced colitis.

**Figure 2 ijms-23-04739-f002:**
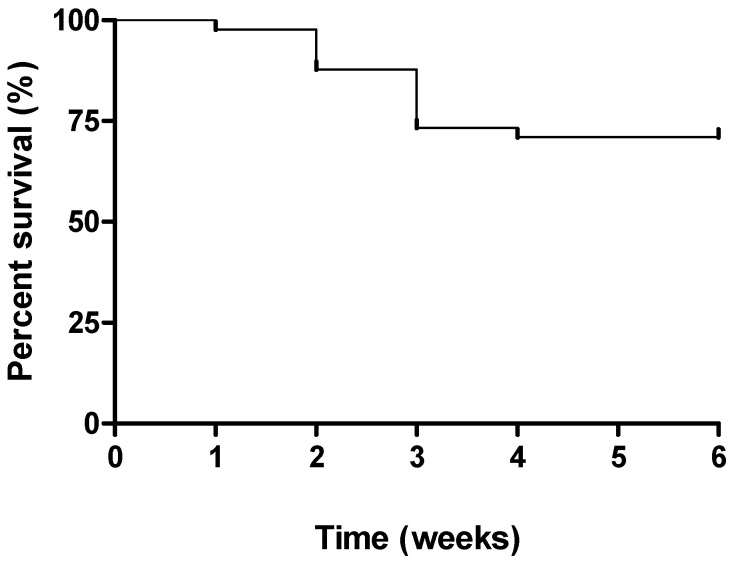
Effect of TNBS-induced colitis on survival rate.

**Figure 3 ijms-23-04739-f003:**
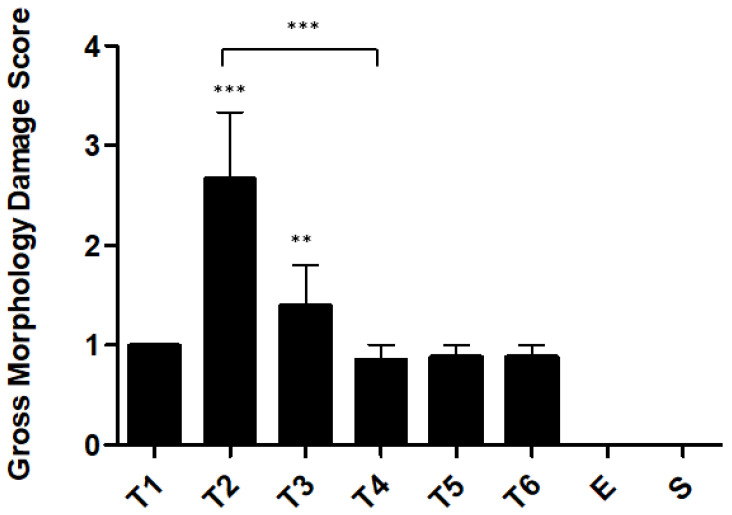
Gross morphology damage score during the development of TNBS-induced colitis. Legend: One-way ANOVA and Tukey’s post hoc test, ** *p* < 0.001 compared with ethanol group, *** *p* < 0.0001 compared with ethanol group or between groups.

**Figure 4 ijms-23-04739-f004:**
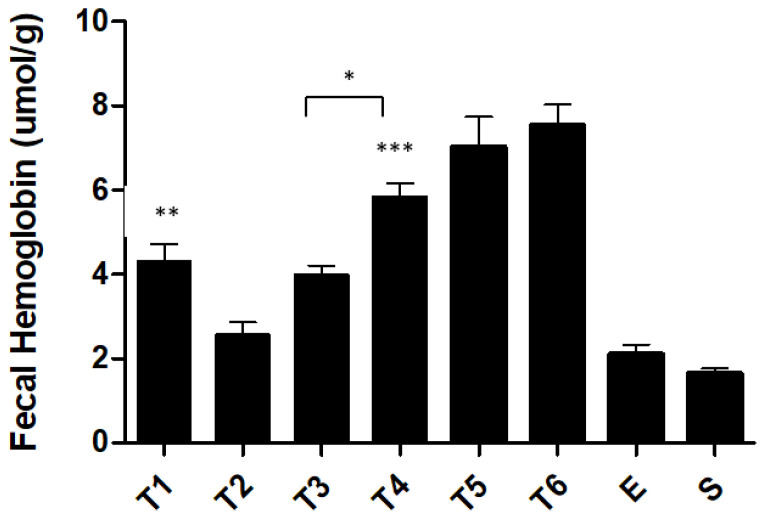
Effect of TNBS-induced colitis on fecal hemoglobin. Legend: One-way ANOVA and Tukey’s post hoc test, * *p* < 0.01 between groups, ** *p* < 0.001 compared with ethanol group, *** *p* < 0.0001 compared with ethanol group.

**Figure 5 ijms-23-04739-f005:**
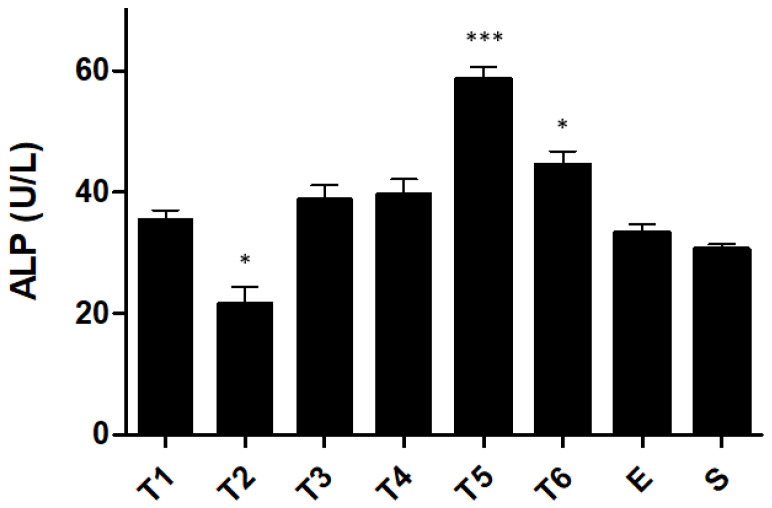
Effect of TNBS-induced colitis on serum total alkaline phosphatase concentration. Legend: One-way ANOVA and Tukey’s post hoc test, * *p* < 0.01 compared with ethanol group, *** *p* < 0.0001 compared with ethanol group.

**Figure 6 ijms-23-04739-f006:**
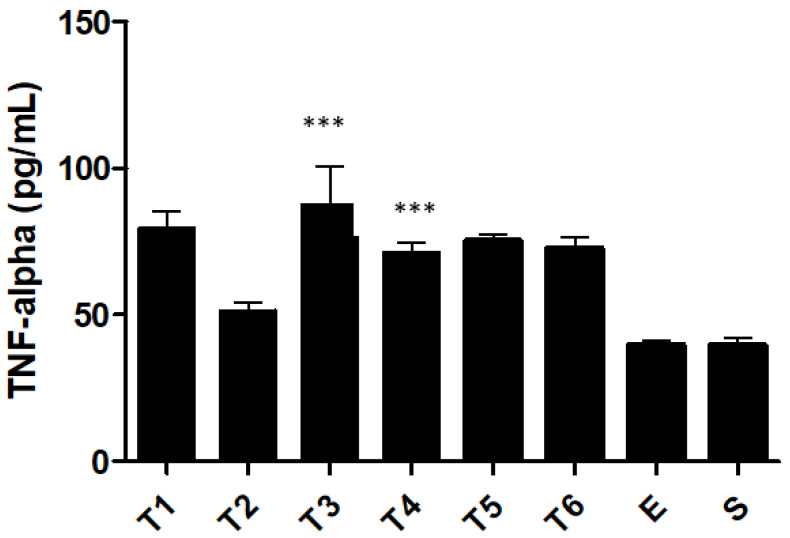
Effect of TNBS-induced colitis on tumor necrosis factor-α concentration. Legend: One-way ANOVA and Tukey’s post hoc test, *** *p* < 0.0001 compared with ethanol group or between groups.

**Figure 7 ijms-23-04739-f007:**
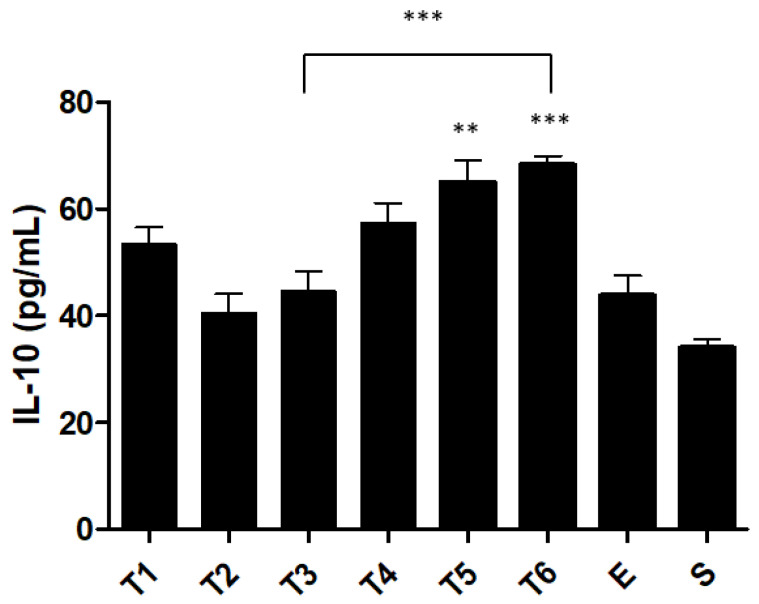
Effect of TNBS-induced colitis on interleukin-10 concentration. Legend: One-way ANOVA and Tukey’s post hoc test, *** *p* < 0.0001 compared with ethanol group or between groups, ** *p* < 0.001 compared with ethanol group.

**Figure 8 ijms-23-04739-f008:**
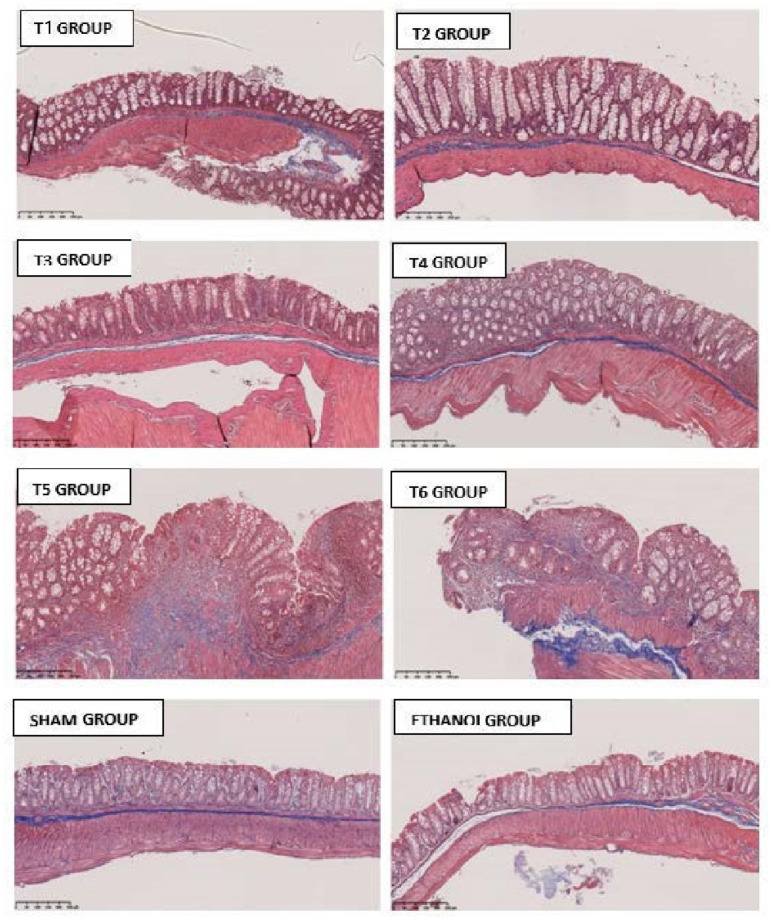
Histopathological analyses of Masson’s trichrome staining 10×.

**Figure 9 ijms-23-04739-f009:**
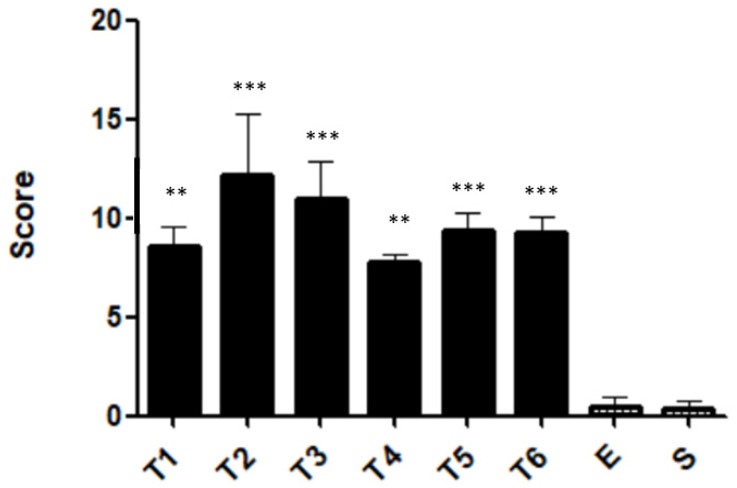
Histopathological score. Legend: One-way ANOVA and Tukey’s post hoc test, *** *p* < 0.0001 compared with ethanol group, ** *p* < 0.001 compared with ethanol group.

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
