# Peer review of "Chronic Experimental Model of TNBS-Induced Colitis to Study Inflammatory Bowel Disease"

_ijms, 2022, doi:10.3390/ijms23094739_

Round 1

Reviewer 1 Report

Silva and staff used TNBS to develop and validate a chronic TNBS-induced colitis in mice, in order to evaluate the effect of new pharmacological approaches for IBD. However, there are several studies have published used this method to induced the same models(PMID:35032589, 35254187,35343079, 35002710), the authors should clarify the novelty of the study and the importance of the model. 

Author Response

Dear Dr,

Thank you for your feedback and all the time you spent working on our manuscript.

The English were revised, and we already improved the manuscript as you can verify.

Response: Animal models of IBD play a pivotal role in the development of new therapeutic approaches to the treatment of IBD and dissect the possible mechanism of action of a particular drug (Randhawa et al. 2014). Chemically induced models are one of the most commonly used to study IBD, because are toxic to colonic cells that generate intense inflammatory response and recruitment of inflammatory cells, representing some of the characteristics observed in human disease (Neurath 2012; Wirtz et al. 2017). Since Morris first described it in 1989, 2,4,6-trinitrobenzenesulfonic acid (TNBS)-induced colitis model has been very popular, because a single rectal administration in rats, mice, guinea pigs, dogs and/or rabbits produces rapid, reliable, and reproducible disease. This animal model, generally accepted as one of the best models for the non-clinical study of a new therapy with the indication for treating or relieving IBD-associated symptoms, is an efficient method, since promotes transmural colitis (Th1-mediated immune response) with severe diarrhea, weight loss, and rectal prolapse, an illness that mimics characteristics of Crohn’s Disease in humans (Motavallian-Naeini et al. 2012; Morris et al. 1989; Wirtz and Neurath 2007).

In the literature, there is no consensus about the induction method and several original articles, like the articles that you suggest, has been published with different ways to induce a chronic model of TNBS-induced colitis, using different doses, number of TNBS administrations, strains, gender, and ages of mice. By this point of view, the main objective of this study is to develop and validate a chronic TNBS-induced colitis in mice, to evaluate the effect of new pharmacological approaches for IBD.

The advantage of this chronic model compared to acute models is that the latter may provide only limited information about the pathogenesis of human IBDs, as the chemical injury to the epithelial barrier leads to self-limiting inflammation rather than chronic disease.

Our research group has developed previous preclinical studies in an acute model of TNBS-induced colitis, testing some pharmacological approaches with new drugs, which presented beneficial effects in the progression and treatment of IBD (Mateus et al. 2017; Mateus et al. 2018; Mateus et al. 2018; Rocha et al., 2015; Direito et al. 2019; Direito et al. 2017). However, IBD is a chronic disease and the development of a standardized and validated induction method for chronic colitis is useful to study new metabolic pathways and, consequently, new pharmacological approaches (Wirtz and Neurath, 2007; Wirtz et al. 2017, Bilsborough et al. 2021).

Sincerely yours,

Prof. Vanessa Alexandra Pinho Mateus, BPharm MSc PhD

Professor of Pharmacology and Pharmacotherapy

Diretor of Pharmacy Degree Course in Lisbon School of Health Technology (Polytechnic Institute of Lisbon)

Member of Coordinating Committee of Health and Technology Research Center (H&TRC)

Reviewer 2 Report

ijms-1680243
Silva et al. describe the utility of a mouse model of TNBS-induced colitis for IBD by altering various conditons. The findings are of interest. However, several points listed below should be considered for publication.
1)    A TNBS-induced colitis model is already widely used for studying IBD-related diseases. Please clearly mention the aim of the study, since the authors have published an important article (ref. #19). The authors must discuss by citing a recent paper (PMID: 33245673).
2)    I guess TNBS-induced colitis is limited in the rectal region and used for studying Crohn’s disease, but not ulcerative colitis. Pathological findings in this study should be carefully described.
3)    In the text, hematoxylin and eosin stain was used for pathological analysis. However, Figure 8 shows Masson’s trichrome staining. Why?
4)    Pathological findings suggesting IBD should be described. The authors showed only mucosal ulcer. Immunohistochemistry of several cytokines can be added.
5)    References are incomplete: there are no authors’ names. Also, they are relatively old.

Author Response

Dear Dr,

Thank you for your feedback and all the time you spent working on our manuscript.

All suggestions have been taken into account and included in the manuscript.

1)    Animal models of IBD play a pivotal role in the development of new therapeutic approaches to the treatment of IBD and dissect the possible mechanism of action of a particular drug (Randhawa et al. 2014). Chemically induced models are one of the most commonly used to study IBD, because are toxic to colonic cells that generate intense inflammatory response and recruitment of inflammatory cells, representing some of the characteristics observed in human disease (Neurath 2012; Wirtz et al. 2017). Since Morris first described it in 1989, 2,4,6-trinitrobenzenesulfonic acid (TNBS)-induced colitis model has been very popular, because a single rectal administration in rats, mice, guinea pigs, dogs and/or rabbits produces rapid, reliable, and reproducible disease. This animal model, generally accepted as one of the best models for the non-clinical study of a new therapy with the indication for treating or relieving IBD-associated symptoms, is an efficient method, since promotes transmural colitis (Th1-mediated immune response) with severe diarrhea, weight loss, and rectal prolapse, an illness that mimics some characteristics of Crohn’s Disease in humans (Motavallian-Naeini et al. 2012; Morris et al. 1989; Wirtz and Neurath 2007).

In the literature, there is no consensus about the induction method and several original articles has been published with different ways to induce a chronic model of TNBS-induced colitis, using different doses, number of TNBS administrations, strains, gender, and ages of mice. By this point of view, the main objective of this study is to develop and validate a chronic TNBS-induced colitis in mice, in order to evaluate the effect of new pharmacological approaches for IBD.

The advantage of this chronic model compared to acute models is that the latter may provide only limited information about the pathogenesis of human IBDs, as the chemical injury to the epithelial barrier leads to self-limiting inflammation rather than chronic disease.

Our research group has developed previous preclinical studies in an acute model of TNBS-induced colitis, testing some pharmacological approaches with new drugs, which presented beneficial effects in the progression and treatment of IBD (Mateus et al. 2017; Mateus et al. 2018; Mateus et al. 2018; Rocha et al., 2015; Direito et al. 2019; Direito et al. 2017). However, IBD is a chronic disease and the development of a standardized and validated induction method for chronic colitis is useful to study new metabolic pathways and, consequently, new pharmacological approaches (Wirtz and Neurath, 2007; Wirtz et al. 2017, Bilsborough et al. 2021).

2)    The 2,4,6-trinitrobenzenesulfonic acid (TNBS)-induced colitis model promotes transmural colitis (Th1-mediated immune response) with severe diarrhea, weight loss, rectal prolapse, an illness that mimics Crohn’s Disease in humans (Motavallian-Naeini et al. 2012; Morris et al. 1989; Wirtz and Neurath 2007). The term “colitis” used in our article refers to the colon inflammation not the Ulcerative Colitis.

3)    We used Masson’s trichrome staining because it is a chronic model, and we wanted to better access the possibility of detecting fibrosis. It was a lapse don’t mention this in the methods. Thanks for the detection.

4)  Sections of distal colon were evaluated based on adapted criteria of Seamons and colleagues (2013). The histopathological score of lesions was partially scored (0–4 increasing severity) with some parameters, namely: (1) presence of tissue loss/necrosis; (2) severity of mucosal epithelial lesion; (3) inflammation; (4) extent 1 - the percentage of intestine affected in any manner and (5) extent 2 - the percentage of intestine affected by the most severe lesion. The colitis severity was calculated by summing the individual lesions and the extent scores, promoting a final colitis score (max score=20) and showed on Figure 9.

5)  It was a lapse. We agree with the reviewer. Thanks for the detection. We will replace the references. Regarding the fact that the references are old, current references were used in the introduction. In the methodology, the references are older, but they are related to the description of techniques. In the discussion, we combined current references with older ones that we considered relevant and with consistent results.

Sincerely yours,

Prof. Vanessa Alexandra Pinho Mateus, BPharm MSc PhD

Professor of Pharmacology and Pharmacotherapy

Diretor of Pharmacy Degree Course in Lisbon School of Health Technology (Polytechnic Institute of Lisbon)

Member of Coordinating Committee of Health and Technology Research Center (H&TRC)

Round 2

Reviewer 2 Report

The revised manuscript has greatly been improved. However,  minor revision is needed for publication.

Please modify the titles (T1  GROUP, etc.) in each photo of Figure 8. They are unclear.